# Cell fate decisions emerge as phages cooperate or compete inside their host

Jimmy T. Trinh[1,2], Tamás Székely[3,4], Qiuyan Shao[1,2], Gábor Balázsi[3,4] & Lanying Zeng[1,2]

The system of the bacterium *Escherichia coli* and its virus, bacteriophage lambda, is paradigmatic for gene regulation in cell-fate development, yet insight about its mechanisms and complexities are limited due to insufficient resolution of study. Here we develop a 4-colour fluorescence reporter system at the single-virus level, combined with computational models to unravel both the interactions between phages and how individual phages determine cellular fates. We find that phages cooperate during lysogenization, compete among each other during lysis, and that confusion between the two pathways occasionally occurs. Additionally, we observe that phage DNAs have fluctuating cellular arrival times and vie for resources to replicate, enabling the interplay during different developmental paths, where each phage genome may make an individual decision. These varied strategies could separate the selection for replication-optimizing beneficial mutations during lysis from sequence diversification during lysogeny, allowing rapid adaptation of phage populations for various environments.

[1] Department of Biochemistry and Biophysics, Texas A&M University, College Station, Texas 77843, USA. [2] Center for Phage Technology, Texas A&M University, College Station, Texas 77843, USA. [3] The Louis and Beatrice Laufer Center for Physical and Quantitative Biology, Stony Brook University, Stony Brook, New York 11794, USA. [4] Department of Biomedical Engineering, Stony Brook University, Stony Brook, New York 11794, USA. Correspondence and requests for materials should be addressed to L.Z. (email: lzeng@tamu.edu).

Decision-making determines the fates of organisms at many levels, from the whole-organism level for multi-cellular beings, where decisions affect how they live and reproduce[1], to the cellular level for all lifeforms, where decisions by single cells can guide development and disease[2,3]. Decisions also have effects at the population level, where perpetuation or extinction hinges on the decision-making of individuals to interact with their neighbours in cooperative and competitive ways to propagate in their environment[4,5]. Therefore, systemic knowledge of cellular decision-making would be instrumental to addressing certain ailments, by potentially manipulating and preventing certain diseases and conditions[6], as well as both understanding the evolutionary history and potentially predicting the evolutionary future of organisms[7,8]. To gain a greater understanding of a complex and ubiquitous concept like decision-making, simple models can be used to simplify and deconstruct its fundamentals[9].

Bacteriophage lambda has served as a paradigm for studying gene regulatory networks, general recombination, bistable switches and other important aspects of cellular life[10–12]. Phage lambda is also a model system for cell-fate developmental decision-making, as it reproduces by infecting its E. coli host, undergoing DNA replication and gene expression, culminating in a decision to develop via either the lytic cycle, where the phage assembles clones of itself and induces cell lysis, or the lysogenic cycle, where the phage genome integrates with the host genome to be replicated by the cell, propagating the phage non-destructively[13]. Although the key genes, genetic circuits, and influential variables affecting the decision have been studied thoroughly over decades[14,15], the underlying mechanisms of how the phage integrates these factors to arrive at cell-fate decisions remain nebulous. However, increased resolution of study in recent years has revealed more deterministic factors and deeper mechanisms[12]. For example, advances in technology allowing for observations at the single-cell and single-phage resolution suggest a reduced role of stochasticity, assigning more importance to pre-existing host variation[16] as well as the existence of independent cell-fate commitments within single cells or 'voting' by infecting phages[17]. This voting phenomenon is particularly interesting as it delves into the interplay between some of the simplest, non-living biological entities, raising intriguing questions about how strands of phage DNA interact with one another and how their decisions shape the evolutionary fitness of the phages, similar to how this process occurs in more complex lifeforms like bacteria and eukaryotes[18,19].

In this study, we synthesize a 4-colour fluorescence system at the single-cell/single-virus/single-viral-DNA level that resolves individual phage votes and interactions to study decision-making in live cells at unprecedented resolution. We also build simple computational models that describe how phages interact as individual DNAs inside cells for lytic/lysogenic fates, to help interpret the data towards mechanisms of decision-making and guide our experimental designs. With this complementary experimental/computational approach, we observe interesting subcellular behaviours, providing new insights into the varied developmental strategies at the level of individual phage DNA, which in turn allows us to understand the effect of this evolutionary strategy on the population. This work also has broader implications as a paradigm for how to quantitatively dissect and understand other regulatory gene networks.

## Results

**Subcellular decision-making assayed using a 4-colour system.**
We achieved higher resolution of phage lambda decision-making via fluorescence imaging of phage gene expression using four fluorescent proteins as our reporters, based on their excellent fluorescence properties and separation on the fluorescence spectrum, mTurquoise2 (ref. 20), mNeongreen[21], mKO2 (ref. 22), and mKate2 (ref. 23). We constructed phages with fluorescent protein genes *mTurquoise2* and *mNeongreen* (denoted blue and green for simplicity), translationally fused to the *λD* gene, which encodes gpD, a capsid decorative protein assembled in >400 copies on the phage head. This enables the visualization of infecting phages and labels progeny phages[24], reporting the progress of the lytic pathway (Fig. 1a). The phages also bear transcriptional fusions of fluorescent protein genes *mKO2* and *mKate2* (denoted yellow and red for simplicity) inserted downstream of the *cI* gene to report lysogeny[25,26], as the *cI* operon is expressed during establishment and maintenance of lysogeny. These transcriptional fusions are expressed as separate proteins to avoid potential interference with CI activity, involving DNA binding and oligomerization[27]. These fluorescent protein genes replace *rexA* and part of *rexB*, genes downstream of *cI*, to preserve the length of the DNA in the operon, as one function of CI is to loop DNA[28,29]. Though *rexB* was suggested to indirectly affect the lytic/lysogenic switch[30], the removal of those genes did not affect lysogenization behaviour (Supplementary Fig. 1). These phages report lytic and lysogenic votes separately: the 'blue phage', with the blue lytic/yellow lysogenic reporter, and the 'green phage', with the green lytic/red lysogenic reporter. Cells infected with both phages report the decisions of both phages, allowing for the visualization of phage voting (Fig. 1a; Supplementary Movie 1). The two phages are distinguishable from the first frame of the movie (Fig. 1b–g, 0 min), and over time, the cells grow, fluorescence develops after cellular decisions occur and the fluorescent proteins mature, allowing us to determine cell fates by the fluorescence signals in the movies (Fig. 1b–g; Supplementary Movies 1 and 2). Interestingly, in lytic cells, the lytic reporter forms foci, and in mixed lytic cells, the two lytic reporters co-localize. We speculate that the foci are centers of phage assembly, and that assembling phages utilize both gpD fusions, which are functionally identical.

As the extensive modifications to the phage genomes may have unknown effects on the phages' lysis/lysogeny decision-making behaviour, we performed bulk-level and single-cell experiments to characterize the lysogenic response of the phages. Bulk lysogenization showed that the two phages have a similar trend of lysogenic frequency versus average phage input (API, ratio of phage titre to host cell concentration) to wild-type (WT) phage (Supplementary Fig. 1a)[31], and single-cell infection movie analyses found that the phages' lysogenic frequency increases with multiplicity of infection (MOI, number of infecting phages per cell), similar to WT phage expectations (Supplementary Fig. 1b). Taken together, these data suggest that the reporter phages behave like WT phage.

To characterize phage interactions, we performed single-cell infection assays with mixed blue and green phages, collected data from all four fluorescence channels at given times for each cell, and normalized every cell's reporter signals to the background fluorescence (Supplementary Fig. 2a–c). We observed crosstalk from the yellow channel into the red, and from the green channel into the yellow. We quantified the level of crosstalk from pure infection movies, using only one of either phage in the absence of the other (Supplementary Fig. 2d,e), and corrected the signals in mixed-phage infection movies by subtracting the crosstalk from the measured signals (Supplementary Fig. 2f,g). For simplicity, we set the highest intensity for a lytic/lysogenic cell in each of the four channels to 100 AU (arbitrary units) and rescaled all cell intensities accordingly. By assigning a cutoff value for each channel, our program is able to assign preliminary cell fates (Supplementary Fig. 2h,i) which are then verified by eye.

**Phages compete in lysis but cooperate in lysogeny.** To determine what types of phage interactions are occurring, we analysed reporter expression patterns in cells infected with both phages (dual-colour infections), and we found that a single lytic/lysogenic reporter may dominate or that both phages' reporters may contribute. For each fate, the scenarios of single phage dominance, where only one phage expresses its gpD fluorescent (lytic) or $cI$ transcriptional (lysogenic) reporter are defined as 'pure lysis/lysogeny signals,' and cases where both phages express their respective reporters are defined as 'mixed lysis/lysogeny signals.' For lytic cells with dual-colour infections, we observed pure lysis at surprisingly high frequency relative to mixed lysis, indicative of possible competitive interactions within the lytic cells. Notably, the number of 'wins' by either phage is similar (45 blue to 41 green), suggesting a scenario where a phage depletes some cellular resource. In contrast to dual-colour lysis, lysogenic cells with dual-colour infections are less likely to show pure red or yellow lysogenic signals (Fig. 2a).

The pure fates we observe may be a combined result of a failed infection (incomplete ejection of phage DNA into the cell), and/or a dark infection (successfully injected phage sheared off before imaging)[17]. To test this possibility, we calculated the predicted pure and mixed signals based on our measured dark and failed infection frequencies (Supplementary Fig. 1c) and compared them with the measurements (Fig. 2a, see Supplementary Methods for calculations). The pure lytic frequency is far above the predictions, indicating that the 'winning' phage must directly or indirectly suppress the other at some time from infection through lysis. Conversely, the pure lysogenic frequency is consistent with the predictions (Fig. 2a), indicating a lack of competition. Additionally, the probability of mixed lysis/lysogeny increases with MOI due to higher probability of successful dual-colour infections. Furthermore,

within the population of pure lytic infections, when one phage outnumbers the other, the cell has a higher chance to report the lytic cycle of the majority phage (Supplementary Fig. 2j,k), indicating that a higher initial phage copy provides an advantage in gene dosage for a phage to dominate lysis by exponential replication. These data suggest that phage interactions vary with cell-fate decisions, where phages commonly coexist in lysogeny, but predominately vie for dominance in lysis.

If phages interact variably during different cell fates, quantifiable differences should be observed when examining the reporters in lytic and lysogenic cells. In lytic cells, the blue or green fluorescence before lysis represents the level of the decorative capsid protein gpD, indicative of phage burst size (Supplementary Fig. 3a,b). For pure infections, the lytic signals of blue and green phages are well fitted to Gaussian distributions with averages of 19.6 and 19.7, respectively. For dual-colour infections of MOI = 2 (one of each phage), we grouped these lytic cells by their fates into pure blue or green lysis (dominating) or mixed lysis. The average lytic signals were 17.9 and 17.0 for blue and green dominating infections, respectively, whereas for mixed lysis, the average blue and green signals were lower, at 13.0 and 10.2, respectively. In the lytic signal distribution, we observed that data points commonly reside near the axes with fewer intermediate points, unlike lysogenic signals, so we devised a measure to usefully differentiate between dominating and mixed reporter signals (Fig. 2b,c). We took the product of the two normalized signals (X and Y values) for each data point, so dominated signals have lower values while mixed signals have higher values, due to being on the plot's edges and middle, respectively. By comparing the lytic and lysogenic distributions, we find that their means (lytic: 265, lysogenic: 603, two-sample Kolmogorov–Smirnov (K–S) test, null hypothesis rejected with $D = 0.41$, $P = 7e\text{-}5$), and their medians (lytic: 30, lysogenic: 159, Mann–Whitney $U$-test, null hypothesis rejected

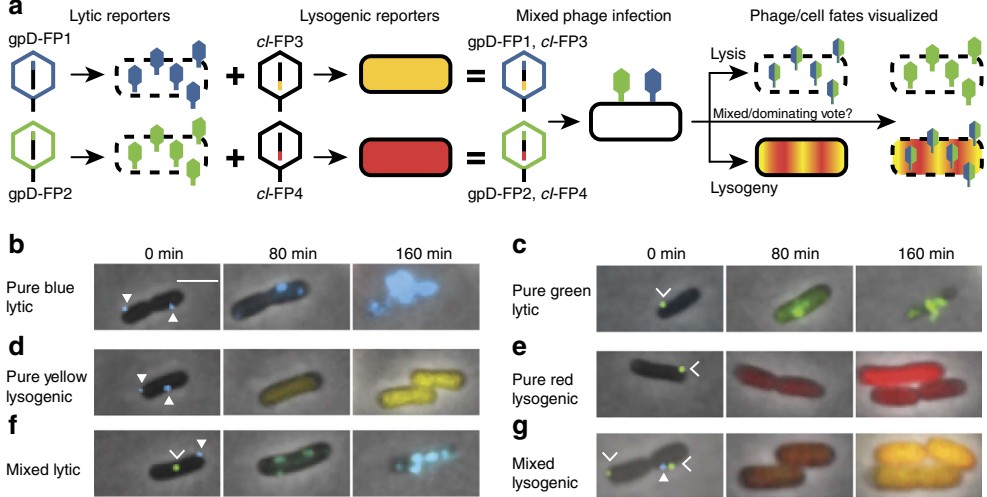

**Figure 1 | Individual phage decisions are visualized using a 4-colour fluorescence reporter system.** (**a**) Lytic reporters are constructed by translationally fusing mTurquoise2 (blue) and mNeongreen (green) to the phage capsid decorative protein, gpD. These phages are visible before infection, and upon lytic decisions, the cells produce progeny phages in the respective colours. Lysogenic reporters are constructed by transcriptionally fusing mKO2 (yellow) and mKate2 (red) to the phage lytic repressor gene, $cI$. Upon lysogenic decisions, cells express the lysogenic reporter colour then grow and divide. Combining the lytic and lysogenic reporters produces two new phages, each with separate decision reporters (blue phage with a blue lytic/yellow lysogenic reporter, and green phage with a green lytic/red lysogenic reporter). Cells infected with both phages (dual-colour infections) show how individual phages make decisions in cells. (**b**–**g**) The overlay images (phase-contrast and fluorescent channels) of representative cells are shown for various cell fates. In the first frames of the movie (0 min), there are filled triangles pointing at blue phages and carets pointing at green phages adsorbed to cells. For lytic cells (**b,c,f**), fluorescence develops over time and forms localized spots in the cell (80 min), followed by cell lysis (160 min). The pure blue and pure green (**b,c**, respectively) lytic cells show only one fluorescence colour, but the mixed lytic cells (**f**) show both blue and green fluorescence, appearing as a cyan colour in the overlay image. For lysogenic cells (**d,e,g**), fluorescence develops uniformly throughout the cells, followed by cell growth and division. The pure yellow and pure red (**d,e**, respectively) lysogenic cells show only one fluorescence colour, but the mixed lysogenic cells (**g**) show both yellow and red fluorescence, appearing as orange in the overlay image. Scale bar, 2 μm.

with $U = 3,701$, $P = 2e\text{-}3$) are significantly different. We also compare the signals over time, where the average lytic fluorescence in pure lysis is greater than in mixed lysis and becomes more significant throughout the infection (Supplementary Fig. 3c,d). Conversely, the fluorescence of lysogenic cells with pure infection (MOI = 2) show similar trends to those of dual-colour infections (one of each phage) (Supplementary Fig. 3e,f) with slightly higher expression for pure infections, which could be due to a difference in the per-phage DNA copy number between mixed and pure infected cells.

The experimental data indicate that phages interact at the DNA level, where lysis and lysogeny exhibit different behaviours, so we built computational models to probe how this might occur. These models separate lysis from lysogeny and simulate what would happen when a cell is infected with one of each phage DNA, where the DNAs would replicate and express reporter proteins (Fig. 2d). We arrived at our parameter set following rigorous parameter testing, confirming that our model is robust (Supplementary Table 3; Supplementary Fig. 7), but would predictably fail to emulate realistic biological processes under incorrect parameter regimes (Supplementary Figs 5 and 6). In both models, one phage DNA enters with a delay, as phage DNA

takes a variable amount of time to translocate into the cell[32]. Also, DNA replication requires an undefined resource to proceed, possibly a polymerase or replisome component[33], which stays bound to the DNA as it replicates[34]. Specific to the lysogenic model, there is also an interaction that converts non-lysogenic DNA into lysogenic DNA, summarizing the biological establishment of lysogeny, where trans-acting CI from lysogens binds all intracellular phage DNAs rendering them as lysogens by repressing other phage genes (see details in Supplementary Methods). Representative simulations show similar distributions compared with experimental lytic and lysogenic cells (Fig. 2e,f). By changing the key parameters of arrival delay and resource level, we can learn about the differences between lytic and lysogenic development, and about the mechanism of domination.

In the lytic model, simulations show that the ratio of DNA levels, and, therefore, lytic reporter levels, of different phage species depend heavily on the relative arrival times of the phage DNAs. At long delays the proteins are often very unbalanced, where one species dominates, and the data points cluster near the axes with few mixed signal points (Fig. 3a,c; Supplementary Fig. 4e,f). With shorter delays, the mixed signal population in the simulations increases. The resource level also affects which

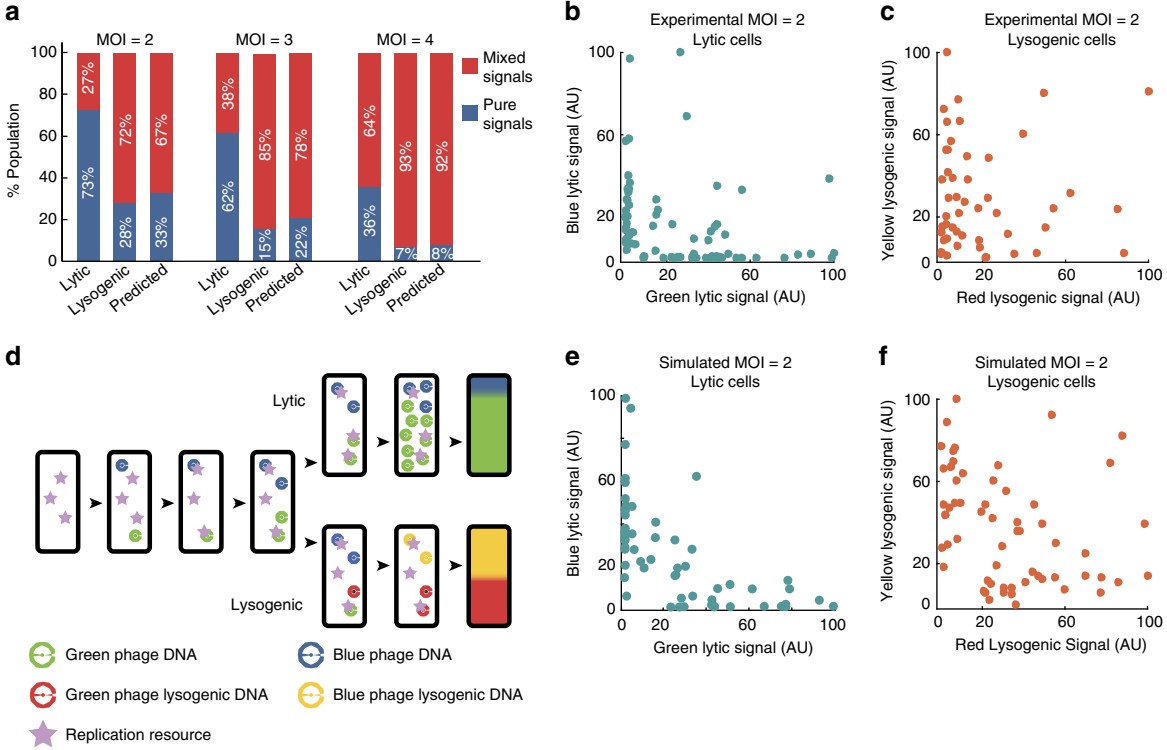

**Figure 2 | Intracellular phages interact competitively during lysis and non-competitively during lysogeny.** (a) Mixed-infected cells at MOI = 2, 3 and 4 are grouped by lytic/lysogenic fates, and furthermore into mixed (both lytic/both lysogenic) and pure (one lytic/lysogenic) fates. The 'predicted' column shows the expected mixed/pure fate populations calculated from observed failed and dark infection frequencies of the blue/green phages. Sample sizes at MOI = 2, 3, 4 are $N = 71$, 45, and 11 (lytic), and $N = 49$, 46, and 30 (lysogenic), respectively. (b,c) Cells infected at MOI = 2, one of each phage, grouped as lytic or lysogenic, regardless of mixed/pure signals. For lytic cells ($N = 71$) (b) we plot their blue/green signals and for lysogenic cells ($N = 49$) (c) their red/yellow signals. Distributions of products for lytic (mean: 265, median: 30) and lysogenic (mean: 603, median: 159) data's $X$ and $Y$ values are significantly different (mean: K–S test, null hypothesis rejected, $D = 0.41$, $P = 7e\text{-}5$; median: Mann–Whitney $U$-test, null hypothesis rejected, $U = 3,701$, $P = 2e\text{-}3$). (d) Diagram of lytic and lysogenic development models for cells infected by one of each blue/green phage shown. DNAs arrive in the cell, bind a resource to replicate, and retain that resource after replication. In lytic cells, DNAs produce a reporter specific to the phage type. The blue/green reporter levels are recorded at the end of each simulation. For lysogenic cells, DNAs can switch into a lysogen after replicating, and lysogens convert DNA into lysogens and produce reporters. Phage arrival times and resource levels are key parameters varied. (e,f) Simulated cells at MOI = 2, resource level = 3, and average arrival delay = 3 replications are shown for the lytic/lysogenic models ($N = 60$ for each), resembling the experimental lytic/lysogenic data. Distributions of mean and median products for lytic (mean: 180, median: 22) and lysogenic (mean: 980, median: 490) data points' $X$ and $Y$ values are significantly different (mean: K–S test, null hypothesis rejected, $D = 0.56$, $P = 3e\text{-}9$; median: Mann–Whitney $U$-test, null hypothesis rejected, $U = 2,457$, $P = 8e\text{-}10$).

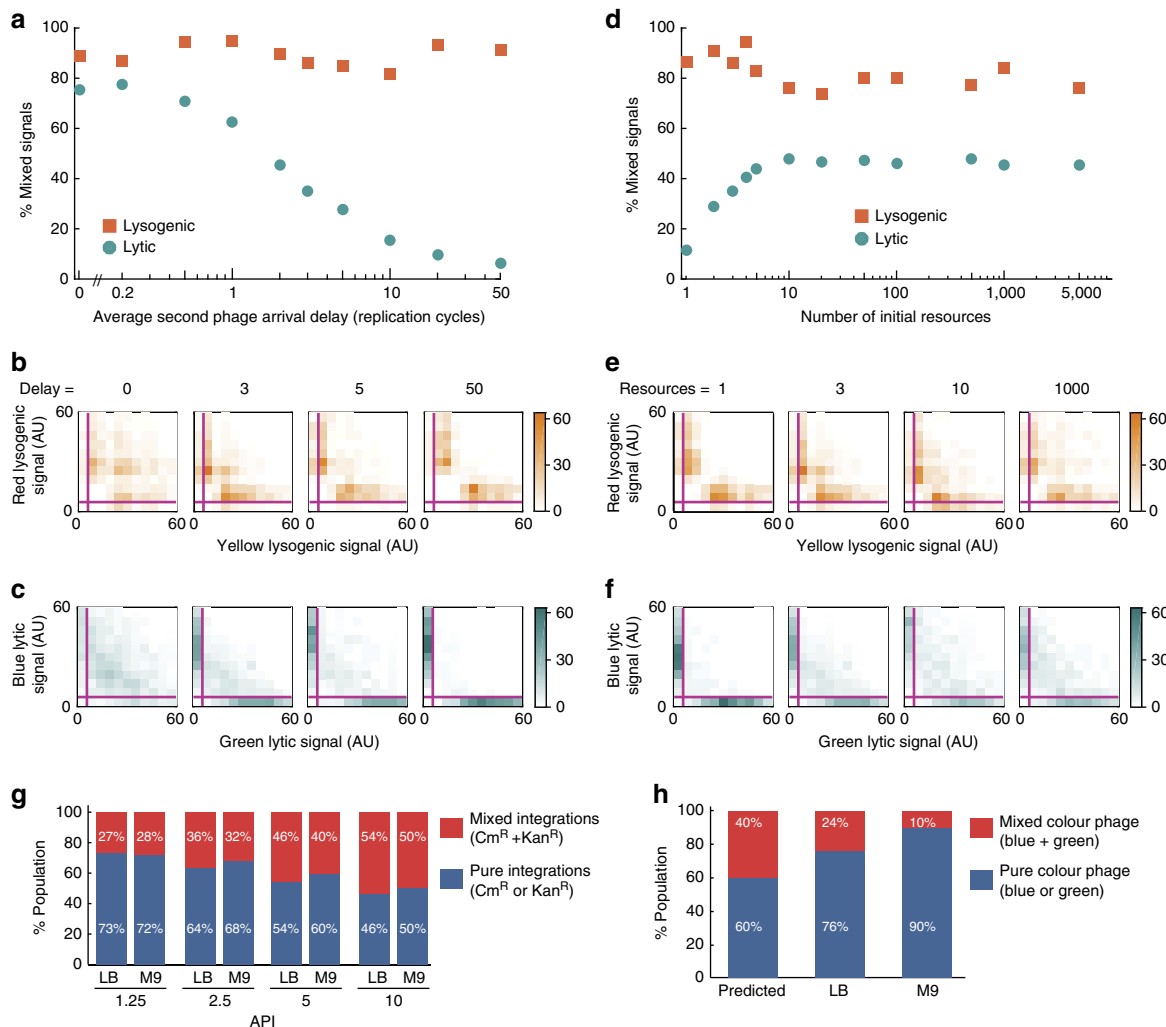

**Figure 3 | Mixed reporter signal frequency decreases with phage DNA arrival time and increases with resource level in lysis but not lysogeny.**
(**a**,**d**) Mixed reporter signals versus average delay time: (**a**) for lytic cells (turquoise circles) decrease with delays whereas for lysogenic cells orange squares) they are relatively constant with respect to delays (fixed resource level = 3, 0 cycles means simultaneous arrival). Mixed reporter signals are plotted against initial resource level in **d** similar to **a** (fixed delay time = three replication cycles). Simulation data (N = 1,000 for all parameter sets) are normalized to the maximum and minimum of each data set and binned (20 bins), where trajectories ending with lytic/lysogenic signal > 5% along both axes are classified as mixed signals for all histograms. Full histograms are shown in Supplementary Fig. 4. (**b**,**c**) Bivariate histograms of lysogenic (**b**) and lytic (**c**) reporter levels from simulations at given arrival delays. Magenta lines represent the pure-mixed threshold (5%) for each reporter. (**e**,**f**) Bivariate histograms of lysogenic (**e**) and lytic (**f**) reporter levels from simulations at given resource levels. Magenta lines represent the pure-mixed threshold (5%) for each reporter. (**g**) Percentage of pure (one antibiotic resistance, Kan[R] or Cm[R]) and mixed lysogens (both antibiotic resistances) from a bulk lysogenization assay using mixed WT phages is plotted as a function of API. The mixed lysogeny increases with API, and is similar between media. (**h**) Percentage of mixed and pure phage progeny from bulk lysis experiment (using the same blue and green phage mixture as in the infection movies) is shown compared with predicted values for different growth media, LB (N = 1,522) and M9 (N = 1,844). Predicted values assume no competition, and mixed population increases in richer LB medium.

species dominates, as with lower resources, the data are shifted towards the axes, whereas mixed signals increase with higher resource levels (Fig. 3d,f; Supplementary Fig. 4a,b). This suggests that competition occurs during DNA replication, where in lysis, one phage has runaway replication to dominate the other. For lysogenic cells, the model predicts that the level of mixed signals is higher than in lysis, and that it is less sensitive to changes in resource level and phage arrival delay (Fig. 3a,b,d,e; Supplementary Fig. 4c,d,g,h). Both models have identical mechanisms for DNA replication, but the lysogenic model predicts less competition for low resources and similar phage DNA numbers for all resources (Supplementary Fig. 4), which is biologically relevant, as lysogenic establishment requires few phage DNA copies and halts DNA replication[15,35]. These models

are, therefore, consistent with our experimental observations regarding different interactions in lysis versus lysogeny, predicting that fast resource depletion is a means of competition.

We next determined the end result of the lytic and lysogenic interactions at the bulk level to support fate-dependent interactions from a different perspective. We performed a bulk lysogenization assay by using a 1:1 mixture of WT phages bearing different antibiotic markers (Kan[R] or Cm[R]) to infect cells at different APIs, to test how mixed phages propagate when finally integrated into lysogens. The results showed a high frequency of double antibiotic resistant cells, indicating mixed phage DNA integration (mixed lysogeny), which increases with API due to the increasing frequency of mixed-phage infections (Fig. 3g). These data are consistent with the lysogenic model's

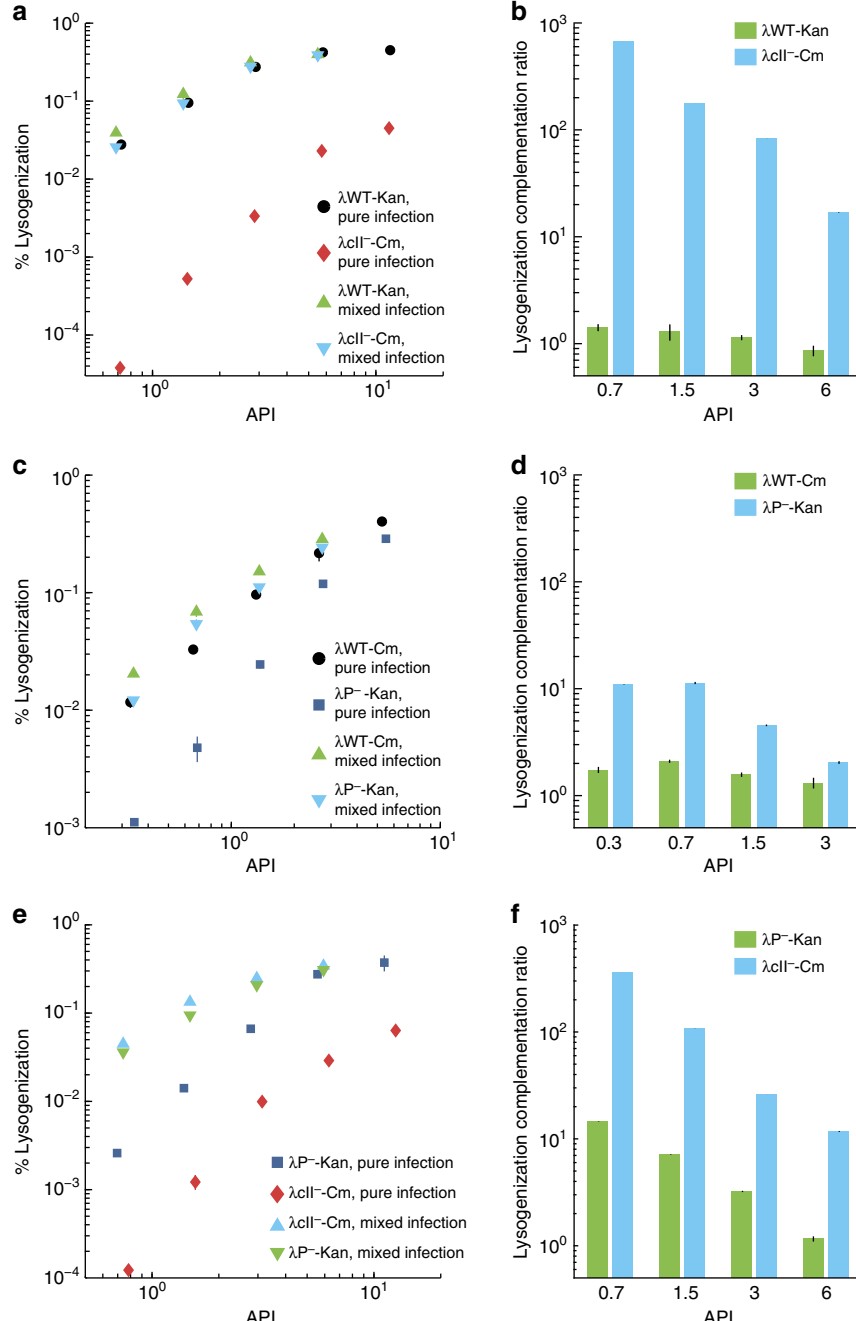

**Figure 4 | Phages cooperate during lysogeny to mutually propagate integration.** (**a,c,e**) Mixed bulk lysogenization using WT phage mixed in a 1:1 ratio with either mutants λcII⁻ (**a**) or λP⁻ (**c**), and 1:1 ratio mixture of λcII⁻ and λP⁻ show complementation of mutant lysogenization defects via co-infection. Lysogenization frequency of pure infections with WT (circles), λcII⁻ (diamonds), and λP⁻ (squares) versus API are plotted on a log–log scale. Total phage integrations from mixed infections including lysogens from pure phage integrations and mixed phage integrations are shown (**a,c**, mutants in down triangles, and WT in up triangles, and **e**, up and down triangles correspond to different mutants) for each API, referring specifically to the number of mutant or WT phages. (**b,d,f**) Quantification of change in lysogenization from pure infections to mixed infections. (**b** corresponds to **a**, **d** to **c** and **f** to **e**). Values are calculated for each API by dividing the % lysogenization in the mixed infection by the % lysogenization in the pure infection; the bar represents the fold change in integration frequency, where '1' is no change. WT shows generally positive changes, and the mutants show increased lysogen frequency substantially. Representative plots are shown for each experiment, which were done with at least two biological replicates consisting of two technical replicates each. Error bars represent ± s.d. of the technical replicates.

prediction of similar DNA numbers, as mixed lysogeny would be favoured in the case of balanced DNA species. This observation holds when the assay is done in either a richer LB or poorer M9 medium, where cells are growing faster or slower, respectively, with similar levels of mixed lysogeny in both media, suggesting that resource limitations in host growth have negligible effect on

lysogenic development and interactions. To test mixed lysis, we infected cells in culture tubes with the same mixture of phages used for the infection movies, forcing the lytic decision in infected cells by incubating at 42 °C, and grouping the resulting phage progeny in the lysate as mixed or pure phages based on microscopy data. The mixed progeny population was far below

the prediction (Fig. 3h), supporting phage competition as observed at the single-cell level. When performing this lytic assay in LB and M9 media, we observed that the richer medium resulted in more mixed progeny, indicating that phages compete over resources involved in host growth during lysis.

The high frequency of mixed lysogeny in the bulk experiment suggests that there may be cooperation during lysogenization. Cooperation can be summarized as individuals in a group interacting non-antagonistically to achieve a mutual benefit for the participants. Cooperating individuals can endow function-deficient mutants with the function that they are lacking, as exemplified by swarming bacteria[5]. We tested such cooperative behaviour for phages by performing the bulk lysogenization assay as described above using lysogenization-defective mutants of phage lambda, $\lambda cII^-$ and $\lambda P^-$ (Supplementary Table 2) in mixed-phage infections, by mixing the mutants with the WT phage and also together, then compared their lysogenization behaviour to pure infections of each strain without mixing[31,36]. We calculated the frequency of mixed integration lysogens (double antibiotic resistance) and pure integration lysogens (single antibiotic resistance) from colony counts on different antibiotic plates. When mixed with WT phages, mutant phage integration frequency increases at all APIs relative to the mutant-only infections (Fig. 4), and WT phage integration frequency generally increases relative to the WT-only infections (Fig. 4b,d). Also of note, the WT phages with different antibiotic markers have the same lysogenization behaviour, and are considered as functionally identical (Fig. 4a,c). When the mutants are mixed together, they mutually complement lysogenization defects, resulting in increased integration frequencies for both phages (Fig. 4e,f). The increases in lysogenic frequency are likely due to the sharing of key lysogenic proteins, and combined with the aforementioned similarity in DNA copies, allow both phages to propagate as lysogens more frequently than was possible in the pure infections. Thus, phages interact cooperatively in lysogeny and show dominating behaviour in lysis, indicating that phages do not behave uniformly in a cell.

**Phages compete via DNA ejection timing and replication**. We next wanted to determine how domination is achieved experimentally, as our model suggests that it occurs from phage DNAs being indirectly blocked from replicating due to resource sequestration. It was reported that significant late gene expression from promoter $P_R$' requires phage DNA replication[37], and impairing phage DNA replication results in phages unable to lyse[38], as in the case of phage DNA replication-deficient mutant $\lambda P^-$ (ref. 31), so any means to inhibit a phage's DNA replication during co-infection could lead to domination. Another potential cause for domination may involve the biochemical kinetics of decision-making, where early commitment by a phage can result in rapid lytic development as to disallow another phage to catch up. Differences in phage DNA ejection timing may vary up to tens of minutes[32], and could facilitate this unsynchronized progression. Phage failure to infect also results in cases of apparent dominant lysis. Thus, to distinguish between infection failure and true domination and to explore the possible mechanisms for domination, we examined specific phage DNA in the cell.

Our strategy to differentially label phage DNA was based on our previous work[39]. SeqA specifically binds fully and hemi-methylated DNA, so a $dam^-$ (methylation deficient) host strain expressing a SeqA-mKO2 fusion was used to specifically label the initial phage DNA (fully methylated) after infection and the first replicated copy of that phage genome (hemi-methylated), but not label unmethylated phage DNA and host DNA[40] (Supplementary Fig. 8a). For this phage DNA reporter experiment, we used a mixture of unmethylated green phage and fully methylated blue phage to infect the reporter strain to visualize the fully methylated phage's DNA when it enters the cell, as a SeqA-mKO2-bound fluorescent focus[39]. The unmethylated phage DNA is not seen but inferred based on the cell fate (Fig. 5a; Supplementary Movie 3). In a typical lytic event, the phage DNA focus is visible in the first frame and divides before lytic reporter activity and lysis (Fig. 5b; Supplementary Movie 3a). We observed that dominating lysis can be due either to the failure of the labelled phage ejecting its DNA into the cytoplasm (Supplementary Fig. 8b,c), or a successful infection (the appearance of a SeqA-mKO2 focus) lacking typical lytic development (Fig. 5c,d; Supplementary Movie 3b,c). The mixed infected population contains a large amount of pure lysis (78%, 69/88 cells), similar to the WT host infections (Fig. 2a). With this DNA reporter system, we are able to divide pure green lysis (56%) into two groups, one resulting from failed infection of the blue phage without SeqA-mKO2 foci (29%) and the other by domination of green phage since the blue phage DNA was successfully ejected with the visible SeqA-mKO2 focus (27%) (Fig. 5e).

We next quantified phage DNA replication in the different lysis groups by tracking the DNA focus, noting whether or not it divides. We found that within the dominating lytic events, the labelled blue phage DNA often does not appear to replicate (Fig. 5c; Supplementary Movie 3b), or the DNA apparently appears late (Fig. 5d; Supplementary Movie 3c). Although replicated DNAs do not always separate using this reporter system, as lytic cells which must have replicated their DNA, occasionally show non-separating foci (12%, 11/91 lytic cells with blue lytic signal), the frequency of non-separating foci during domination (62%, 15/24 dominated lytic cells with non-separating foci) is much higher, indicative of non-replicated phage DNA (Fig. 5f). These findings were reciprocated in experiments with switched phage methylation states (Fig. 5e; Supplementary Fig. 8d–f; Supplementary Movie 3d,e). The data suggest that earlier infection timing provides advantage for phages to compete during the subsequent exponential phage DNA replication, but does not conclude whether these factors are related, as predicted by our modelling.

**Confusion in mixed voting results in delayed lysis**. The different phage interactions in a cell support a prediction from the voting model: mixed lytic and lysogenic phage voting within a cell[17]. We observed these events in our experiments (7%, 67/1,006 infected cells), which were classified in the lytic category (Fig. 6a,b). Mixed voting was designated when cells lysing with both phage colours also expressed one or both lysogenic reporters (Fig. 6a top row; Supplementary Fig. 9a; Supplementary Movie 4a, denoted cross-phage mixed vote), or cells lysing with one phage colour and showing the lysogenic reporter of the same phage (red in green lysis, or yellow in blue lysis) (Fig. 6a bottom two rows; Supplementary Fig. 9a; Supplementary Movie 4b,c, denoted same-phage type mixed vote). Interestingly, same-phage type mixed voting can occur when the cell is apparently only infected by one phage (7/20 mixed vote green, and 12/35 mixed vote blue at MOI = 1), suggesting that individual phage DNA, even replicated genomes, can interact with each other and independently commit to either lytic or lysogenic pathways in the same cell. In addition, mixed voting frequency increases with MOI (Supplementary Fig. 9b), similar to lysogenization frequency increasing with MOI[17,31].

Mixed voting occurs regularly in lytic cells (14%, 67/481 lytic cells), so we asked how this interaction affects lytic progression to determine its purpose. Mixed voting cells exhibit lysogenic

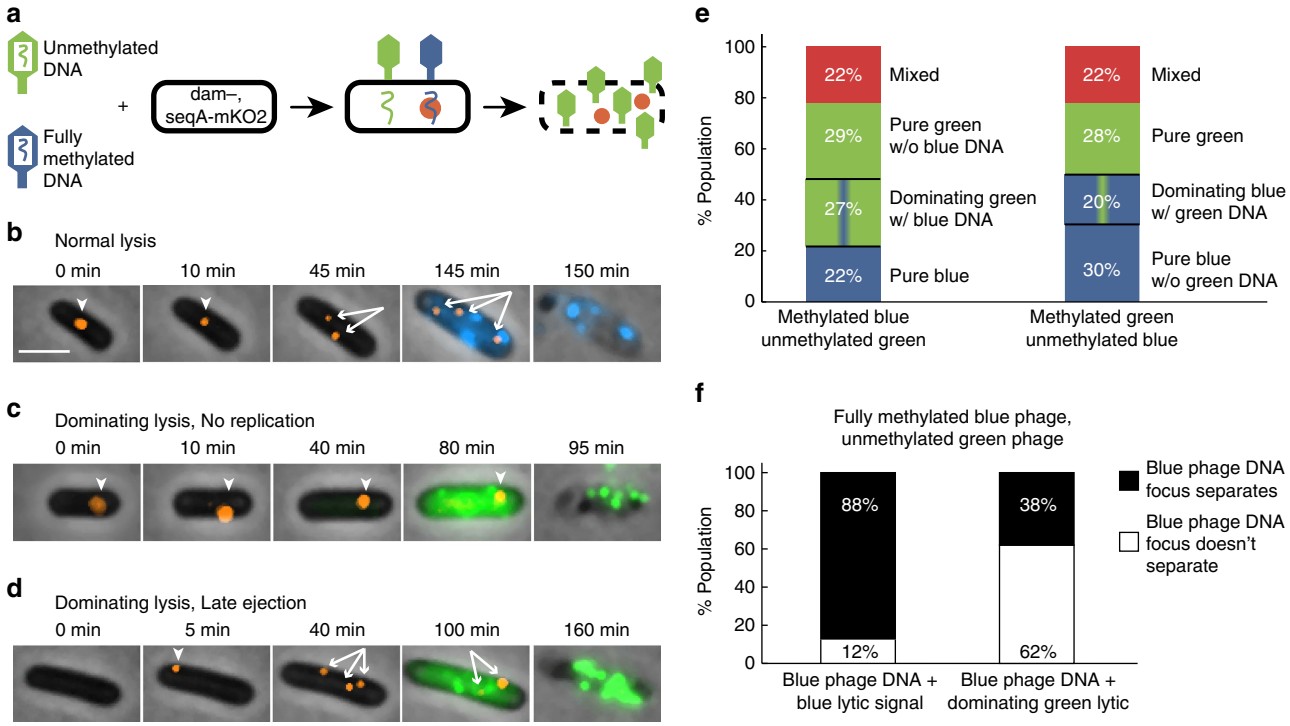

**Figure 5 | Dominating lysis results from phage competition during DNA ejection and replication.** (**a**) In *dam⁻ seqA-mKO2* cells, the fully methylated phage DNA is bound by SeqA-mKO2 forming a focus. This reporter can show when DNA replicates once, producing two hemi-methylated DNAs bound by SeqA-mKO2. Dominating lysis occurs when unmethylated phage shows pure lysis with intracellular fully methylated phage DNA. (**b–d**) Overlays showing different lysis types (blue/green phage is fully/unmethylated). Orange foci represent the first cellular DNA observed (arrowheads). After apparent DNA replication, multiple foci appear (branched arrows). (**b**) Normal lysis. Phage DNA seen at 0 min, two DNA foci at 45 min, then cell develops blue fluorescence and lyses. (**c**) Dominating lysis. DNA focus seen at 0 min, but the focus does not apparently split, and cell lyses with only green fluorescence. (**d**) Dominating lysis. DNA focus is absent at 0 min, appearing at 5 min. DNA focus divides, but green fluorescence accumulates, not blue. (**e**) Different lysis groups in dual-colour infection of fully methylated blue/unmethylated green phages (left, N = 88) and of fully methylated green/unmethylated blue phages (right, N = 85). Left, when blue phage DNA is labelled, pure green lysis (56%) is divided into pure green lacking blue DNA (failed blue phage infection, 29%), and dominating green with blue DNA (27%). Right, when green phage DNA is labelled, pure blue lysis (50%) is divided into pure blue without green DNA (failed green phage infection, 28%), and dominating blue with green DNA (20%). (**f**) Lytic cells with a focus (blue phage DNA) in mixed-phage infections (the blue/green phage is fully/un-methylated) are divided: lysis with blue signal (left, N = 91) and dominating green (right, N = 24). Within each lytic group, the frequency of the DNA focus separating into multiple foci is plotted. As lysis requires DNA replication, the 12% non-separating group in the blue lytic group represents basal failure of reporting DNA replication (left). The 62% non-separating group in the dominating green lytic group (right) is higher than basal failure. Of the dominated DNA that does divide, 6/9 showed late ejection. Scale bar, 2 μm.

development, which is expected to inhibit lytic development compared with normal lytic cells, where we used fluorescence as a proxy for lytic development. The average lytic signal for pure lytic cells accumulates, followed by a decrease over time, due to the lysis of infected cells (Supplementary Fig. 3c,d, green squares). Cells classified as mixed voting show lower average level of lytic expression through 100 min after infection, but afterwards show a higher maximum. For mixed voting cells, slower signal accumulation could be due to a lysogenic phage DNA population interfering with lysis, but being unable to repress the entirety of intracellular phage DNA. This would delay the production of lytic signal and cause mixed voting cells to lyse later than normal lytic cells. The average lysis times for each group were quantified from the distribution of lysed cells per time point: mixed voting blue and green lytic = 147 and 151 min, respectively; pure blue, green and mixed lytic = 114, 114, and 115 min, respectively (Fig. 6c). The mixed voting cells take longer to complete lysis, suggesting that the confusion between decisions negatively impacts phage propagation. When the mixture of phages infected the WT host, we observed that lytic development inevitably led to lysis, but when the mixture of fully methylated and unmethylated phage infected the DNA reporter host, there were rare cases of mixed voting (<1%, 5/1,371 cells) where lytic development gave way to strong lysogenic expression followed by cell division instead (Fig. 6d; Supplementary Movie 5), exemplifying a possible function of mixed voting.

## Discussion

In this work, we developed a 4-colour fluorescence system to study phage decision-making at higher resolution by integrating different reporters into phage genomes, distinguishing between two different phages to determine individual lytic/lysogenic decisions within cells. This system can serve as a platform to explore phage behaviours and interactions under varied growth conditions, host backgrounds, phage mutants, and also using different phages, to characterize variables related to cellular decision-making. The combination of these fluorescent proteins may be used in other systems requiring labelling of multiple pathways, expanding the resolution of study for other models.

We found that phage interactions in the cell are either cooperative or competitive depending on the decision-making of the phages. During lysis, ~100 progeny are assembled and packaged regardless of MOI, due to the timing of cell lysis by holin activity[41], so dominating phages win a larger share of progeny. In lysis, the phage propagates itself rapidly,

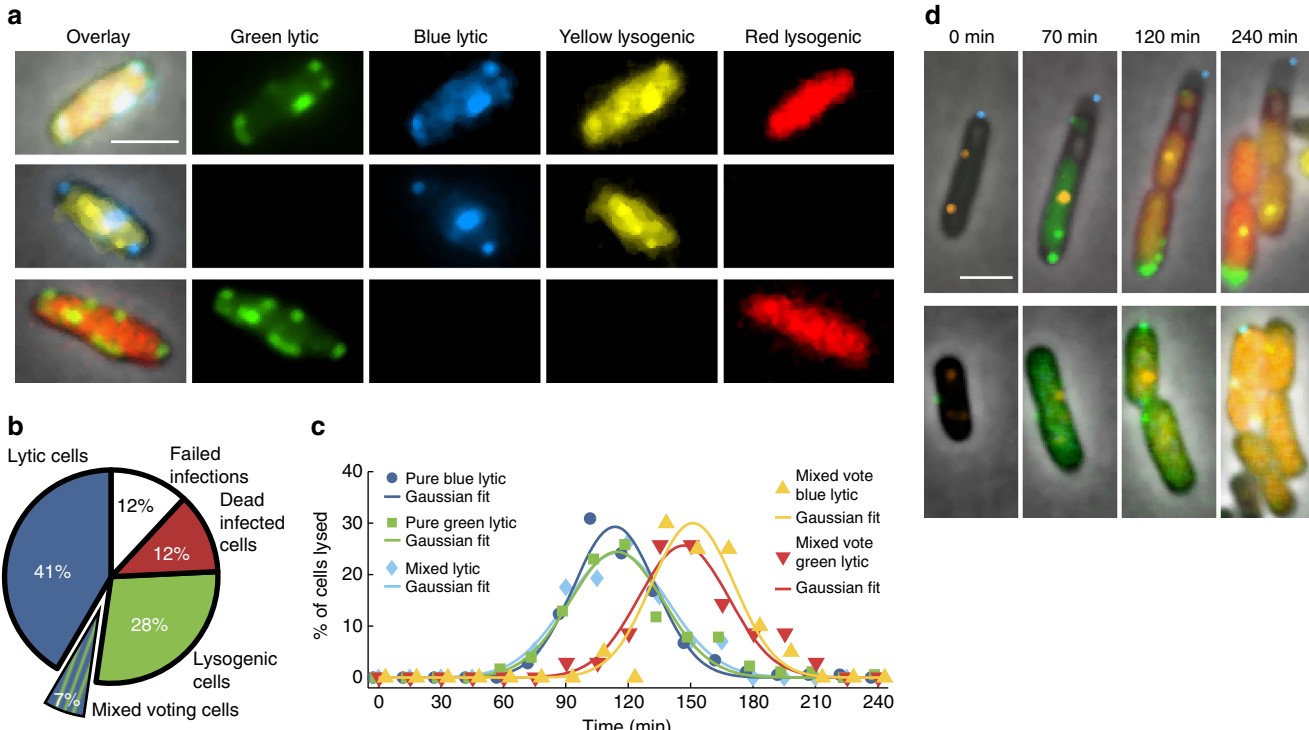

**Figure 6 | Mixed voting for fates occurs between phage DNAs in a cell and delays lytic development.** (**a**) Examples of three different mixed voting cells including overlay images and their component fluorescence channels. The top row shows a 'cross-phage mixed voting' cell with green and blue lytic signals and with red and yellow lysogenic signals. The middle and bottom rows show examples of 'same-phage type mixed voting', as both lytic and lysogenic reporters can be found on the same phage. (**b**) A pie chart is shown comparing the different fates of all cells in this study (1,006 infected cells). Seven per cent are mixed voting cells resulting in lysis. (**c**) Lytic cells were divided into different groups: pure blue ($N = 178$) (circles), pure green ($N = 179$) (squares), mixed lytic ($N = 57$) (diamonds), mixed voting blue lytic ($N = 35$) (up triangles) and mixed voting green lytic ($N = 20$) (down triangles). The number of cells lysed since the previous time point, as a percentage of the total group, is plotted with time. Each group's distribution is well fitted to a Gaussian distribution (lines), with an average lysis time of 114 min (pure blue), 114 min (pure green), 115 min (mixed lytic), 147 min (mixed voting blue) and 151 min (mixed voting green). (**d**) Example cells show lytic development yet lysogenic development and cell division using phage DNA reporter cells and fully/unmethylated phages. In the top and bottom rows the green and blue phages are methylated, respectively, with orange foci representing the ejected/replicated DNA. Green lytic signal builds up, but with time, the lysogenic reporter expression occurs (red or orange, the DNA reporter shares the same fluorescent protein as one lysogenic reporter) and the cell divides multiple times during the course of the movie, while the lytic signal ceases to accumulate. Scale bars, 2 μm.

so competition is beneficial for phages with favourable genetics specifically in conditions promoting lysis. Our model predicts that phage DNA replication is a central point of contention for phage interactions, highlighting the importance of DNA replication in the infection, even though the process is often overlooked. Because phages can replicate their genomes in 2–3 min (refs 42,43), asynchronous DNA ejection timing can account for variation between phages' DNA copy numbers and, therefore, gene expression at given times[44], providing a basis for competition. It is also known that a phage can be starved during an infection, rendering it dormant in a state called pseudolysogeny[45], which appears similar to a dominated phage's behaviour. As lysis requires extensive phage DNA replication, a phage could siphon essential host proteins for phage DNA replication, such as *E. coli* Pol III holoenzymes or DnaB hexamers, both estimated to be low in number[33,46], preventing other phage DNA from developing. Phage genomes were also suggested to inherit replication complexes during replication, thus sequestering replisomes from other DNA[34]. Additionally, the critical late gene regulator, *Q*, was reported to be preferentially *cis*-acting[47,48], which would facilitate competition. For lysogenic cells, cooperation is the typical outcome, reported at the level of transcription and phage integration. This may be attributed to the lower phage DNA copy number requirement for lysogeny,

as $\lambda P^-$, deficient in phage DNA replication, can lysogenize[31]. DNA replication, and thus competition, does normally occur in lysogeny, but lysogenic establishment halts DNA replication, so there is balance in DNA numbers. Our data suggests multiple prophages propagate frequently in lysogens, which would enhance genetic variance, allowing mutants arising from DNA replication as well as co-infecting lysogenization defective mutants to persist in the lysogenic state. The phage interactions and specific gene functions may have evolved to strategically sustain the viral population: when conditions are ripe for propagation, competition ensues, driving the selection of genes favourable for fast reproduction within that environment. However, when conditions favour lysogenization, typically in poor, host-limited conditions, phages cooperate to integrate their genes as lysogens, and replicate in the absence of selection. The lack of direct selection on phages enables their diversification by genetic drift[49], perhaps aiding subsequent adaptation to new environments (Fig. 7). Nevertheless, while selection on phages is absent during lysogenic propagation, selection exists at the host level: dormant lysogens with maladaptive mutations for the host will probably disappear from the population.

Phage voting is independent at the phage DNA level, as studies have observed infected cells concurrently displaying lysogenic and lytic reporter activities, where lysis overcomes lysogeny[17,50].

**Figure 7 | Strategic DNA level phage interactions during development increase evolutionary fitness.** Individual phage DNAs make decisions to develop via lytic or lysogenic pathways, and interact with each other based on the decisions. During lysogeny, DNA replication is limited and favours mixed integrations of phage genomes. This cooperation may help diversify lysogens with different phage DNA to produce varied phages when induced later. Varied phages can be beneficial if the cells move to unknown environments. During lysis, extensive DNA replication results in resource competition, which typically favours a single phage type. Competition during lysis allows good genetics to propagate. If conditions favour lysis, dominating phages can spread quicker and more thoroughly. Different phage DNAs may choose different fates, which delays lytic progress. Confusion during development is non-optimal for propagating quickly, but if the delay allows survival in some rare situations, the tradeoff may increase the overall fitness of the phage.

Our data suggest that during mixed voting, a subpopulation of phage DNA commits to lysogeny, but another subpopulation is unrepressed and commits to lysis. Our work is in agreement with previous work suggesting that steps in lytic development can kill or lyse the cell, overriding lysogeny, such as excessive phage DNA replication[51], or expression of Q-mediated lytic products[52]. That this mixed voting occurs even in MOI = 1 infections is evidence that each replicated phage DNA is making decisions and interactions to replicate and sustain itself. By tracking single phage DNAs, we have found that labelled DNAs occasionally move as if they are 'trapped' in the cytoplasm, suggesting that phage genomes may occupy distinct intracellular locations, with potentially different microenvironments favouring different decisions[39]. We observed in $dam^-$ cells, deficient in DNA repair and thereby prone to mutation[53], using mixed fully methylated and unmethylated phages, that lytic development could be overcome by lysogenic activity, and develop similarly to normal lysogens (Fig. 6d; Supplementary Movie 5), raising the possibility that mixed voting is a fail-safe against certain host/phage mutations, where lysogeny would be somehow preferable to lysis. The phage DNAs can develop in both pathways, and sometimes commit to different decisions, where specific intracellular conditions determine the final outcome (Fig. 7) This scenario has an unknown cause, possibly due to disparate, transient local environments, and may have evolved as an insurance strategy to save some phages from niche situations, with a delayed lysis tradeoff.

The results of this study illustrate how we may discover new details and revisit an established system from a new perspective by expanding the resolution of study. The phage interactions, and the physiologically relevant factors potentially responsible for them, like phage DNA replication and asynchronous infection timing, are crucial aspects to account for when modelling the system. Yet, these factors were not typically considered in previous models. Our approach was to incorporate these new effects into simple models instead of adding them to existing detailed models and increasing their complexity further. The insights we gained demonstrate how improved modelling of natural systems is not always synonymous with increasing the complexity of models. Obtaining new fundamental parameters and including them into simple models can actually facilitate the generation of new insights compared with their incorporation into complex models. Moreover, these simple models of decision-making can also relate the individual phage's behaviour to its evolutionary strategy. Technology allows us to push our limits when studying systems to observe the previously unobservable, allowing biological models to be further revisited, improved and understood.

## Methods

**Strains and plasmids.** The host strain for normal phage movies is MG1655. The host for phage DNA tracking movies, MG1655 seqA-mKO2/mKate2 $Cm^R$ $\Delta dam$::$Kan^R$ strain was produced by recombineering the fluorescent fusion protein with the chloramphenicol resistance cassette fused to the native seqA gene using the λ red system[54] into MG1655, and then the same system was used on the resulting cell to replace the dam gene with the kanamycin resistance cassette. For antibiotic selection, 10 µg ml$^{-1}$ of Cm, 50 µg ml$^{-1}$ of Kan, 100 µg ml$^{-1}$ of Amp and 10 µg ml$^{-1}$ of Tet were used as appropriate.

New reporter phages are λD-mTurquoise2 $cI_{857}$-mKO2 bor::$Cm^R$ and λD-mNeongreen $cI_{857}$-mKate2 bor::$Cm^R$ (blue and green phages, respectively). With the exception of the $\lambda cI^-$ phage, all of the phages and their lysogens have the $cI_{857}$ allele[55], which does not undergo auto-cleavage by the host SOS system[56] and is temperature sensitive (lysogeny is non-permissive at 37 °C).

All phages with bor::$Kan^R$ were produced through recombination by titering onto cells with plasmid pER157 (gift of Ryland Young) , and phages with bor::$Cm^R$ were produced through recombination by titering onto cells with plasmid pZA3-R-Cam-cos[57].

All bacterial strains and plasmids used are listed in Supplementary Table 1, and phages used in Supplementary Table 2.

**Phage strain construction.** The fluorescent proteins in the recombination plasmids, pBR322-D-mTurquoise2/mNeongreen-E and pBR322-$cI_{857}$-mKO2/ mKate2-partRexB, were cloned from plasmids with each individual gene: mTurquoise2 (received from Stanislav Vitha as a gift from Theodorus W. J. Gadella), mNeongreen (from Allele Biotechnology), mKO2 (received as a gift from Michael Davidson (Addgene plasmid # 54625)) and mKate2 (received as a gift from Anna Planas and Tomas Santalucia (Addgene plasmid # 48345)). WT lambda phages were recombineered into the green and blue reporter phages via infecting host cells bearing the recombination plasmids. Detailed protocols can be found in the Supplementary Methods.

**Single-cell infection assay.** Both phages (λLZ1367 and λLZ1373) were purified (detailed protocol is described in the Supplementary Methods) then diluted to the same titre and mixed together at a 1:1 ratio to generate the phage mixture for infection. Infection was done as previously described[17], using M9 minimal medium as the growth medium in order to have optimal fluorescence signals. The phages used for infection were gpD-mosaic with a mixture of WT gpD and gpD fluorescent fusions, to avoid capsid instability due to assembly of too many copies of the gpD fluorescent fusion proteins.

Briefly, a 1 ml overnight culture of host MG1655, grown at 37 °C (265 r.p.m. shaking) in M9 + 0.4% maltose (M9M), was diluted 1:100 into M9M (50 µl of overnight culture into 5 ml of M9M), and grown at 37 °C (265 r.p.m. shaking) until $OD_{600} \sim 0.4$. Once grown, 1 ml of the culture was centrifuged (4 °C, 2,000g, 4 min), supernatant was discarded, and cells were resuspended in 150 µl of cold M9M. Twenty microlitres of the phage mixture was then mixed with 20 µl of the resuspended MG1655 culture to infect. The infection mixture was left on ice for 30 min, then 80 µl of cold M9M was added to dilute the infection mixture (pipette tips cut for wider opening), and gently mixed by tapping, then moved to a 35 °C

water bath for 5 min, then 1 μl of the mixture was placed (pipette tips cut for wider opening) onto a 1.5% agarose pad of M9M (prepared by microwaving 0.09 g of agarose (Fisher Scientific) with 6 ml of M9 media, then adding 120 μl of 20% maltose to the molten agarose) resting on a small No.1 coverslip (18 × 18 mm, Fisher Scientific) until visibly dry (~1 min), then covered by a large No. 1 coverslip (24 × 50 mm, Fisher Scientific), and then moved to the microscope for time-lapse imaging, where the time = 0 is set to the first time-lapse image taken, which typically will be ~15–20 min after cells are initially placed in the 35 °C water bath.

For the phage DNA reporter movies the same conditions were used but with the reporter host strain (MG1655 $seqA$-mKO2 $\Delta dam::Kan^R$) infected with a 1:1 mixture of mixed methylation blue/green phages (λLZ1379 + λLZ1380 or λLZ1378 + λLZ1381). Similarly, for pure infection movies, a single purified phage is used instead of a mixture.

**Microscopy imaging.** Imaging was performed on a Nikon Eclipse Ti inverted epifluorescence microscope using a $100 \times$ objective (Plan Fluo, NA 1.40, oil immersion) with a $2.5 \times$ TV relay lens, using a mercury lamp as the light source (X-Cite 200DC, Excelitas Technologies), within a cage incubator (InVivo Scientific) at 30 °C, and acquired using a cooled EMCCD (electron multiplying charge-coupled device) camera (iXon3 897, Andor, Belfast, United Kingdom). For a typical movie, 8 or 16 stages were selected where cells were well separated but plentiful. The software images each stage through each filter sequentially for each time point before moving to the next stage. The cells were imaged under the phase-contrast and four fluorescent filter cubes. The fluorescent filters used in the study were as follows (Xnm, Yex [bandwidth] excitation filter/dichroic beamsplitter wavelength/Xnm, Yem [bandwidth] emission filter/company, product #): blue (436 nm, 20ex/455 nm/480 nm, 40em/Nikon, 96,361), green (490 nm, 20ex/505 nm/525 nm, 30em/Chroma, custom 49,308), yellow (539 nm, 21ex/556 nm/576 nm, 31em/Chroma, 49,309) and red (592 nm, 21ex/610 nm/630 nm, 30em/Chroma, 49,310). The first frame of the movie imaged cells with z-stacks of $\pm 1.2$ μm, 0.3 μm each step, under the blue and green filters, to visualize infecting phages surrounding the cells. For this first frame, the images are acquired in this order (with exposures): phase-contrast (100 ms), red (200 ms), yellow (100 ms), blue (200 ms) and green (200 ms), and the phase-contrast, red and yellow channels are only taken at the focal plane. The time-lapse movies were then taken every 5 min without z-stacks for ~4 h after all cell decisions were resolved. For the time-lapse portion, the images are acquired in this order (with exposures): phase-contrast (100 ms), red (1 s), yellow (100 ms), blue (40 ms) and green (40 ms).

For the DNA reporter movies, z-stacks (focal plane and +0.4 μm) are taken in the DNA reporter channel throughout the whole movie to track the DNA focus. The first frame of the movie imaged cells with nine z-stacks of $\pm 1.2$ μm, 0.3 μm each, under the blue and green filters, to visualize infecting phages surrounding the cells. For the first frame, the images are acquired in this order (with exposures): phase-contrast (100 ms), red (200 s), yellow (200 ms), blue (200 ms) and green (200 ms). For time-lapse portion, the images are acquired in this order (with exposures): phase-contrast (100 ms), red (1 s), yellow (200 ms), blue (40 ms) and green (40 ms).

When presenting microscopy images in our figures and movies, uniform contrast settings are applied for each separate channel throughout the entire figure subpanel or movie.

**Analysis of time-lapse movies.** Movie images were analysed using the cell recognition program Schnitzcell (gift of Michael Elowitz, California Institute of Technology) and homemade script in Matlab. Cell lineages were determined from the phase-contrast images. The fluorescent signal was normalized to the background, corrected by the crosstalk and then scaled for further analysis.

**Bulk lysogenization assay.** The reporter phages, λD-mTurquoise2 $cI_{857}$-mKO2 $bor::Cm^R$ (blue phage) and λD-mNeongreen $cI_{857}$-mKate2 $bor::Cm^R$ (green phage), and WT phage, λ$cI_{857}$ $bor::Cm^R$ (λWT-$Cm$), were diluted using SM buffer to an approximate API of ~10 based on previously known titres. Twofold dilutions done in duplicate were then made to reach a final concentration of $2^{-6}$ and the seven samples for each phage were kept on ice until infection. Host E. coli MG1655 from an overnight culture was diluted 1:1,000 into LB + 0.2% maltose + 10 mM MgSO$_4$ LB + maltose + MgSO$_4$ (LBMM) (10 μl of overnight into 10 ml LBMM) and grown at 37 °C, 265 r.p.m. until an OD$_{600}$ of ~0.4, where it was then concentrated through centrifugation (4 °C, 10 min, 2,000g), then resuspended in fresh LB + 10 mM MgSO$_4$ (LBM) using one-tenth of the original volume (cells concentrated 10 ×). Twenty microlitres of the cell suspension and 20 μl of each phage dilution were mixed to make an infection mixture, and the infection mixtures were kept on ice for 30 min. The infection mixtures were then transferred to a 35 °C water bath for 5 min, and then 10 μl of each infection mixture was diluted separately into 1 ml of pre-warmed LB + 0.2% glucose + 10 mM MgSO$_4$ LB + glucose + MgSO$_4$ (LBGM) in a 30 °C, 265 r.p.m. water shaker for 45 min. The new infection mixture was then transferred to ice and diluted with cold PBS (Lonza) appropriately to allow ~100–500 colonies to grow, and then 100 μl of each dilution was spread onto LB + $Cm$ plates and incubated overnight at 30 °C to select for lysogens. Lysogen counts were determined by counting the number of $Cm^R$ colonies. Pre-infection phage and bacteria concentrations were measured using

standard plate assays, titre using LE392 as the indicator strain for phage and plating on LB for bacteria. The lysogenization probability was plotted as a function of API on a log-log scale (Supplementary Fig. 1a). It was found that the reporter phages exhibit the same API-dependent lysogenization response as wild type.

Different types of phage integrations were determined using the same protocol as above with different phage solutions. A 1:1 mixture of WT phages bearing a $Kan^R$ marker and WT phages bearing a $Cm^R$ marker (λWT-$Kan$ + λWT-$Cm$) was diluted with SM buffer to API ~10, then duplicate twofold serial dilutions using SM were done to reach API ~1. In the final plating step, for each lysogen culture, 100 μl was plated on LB + $Kan$, LB + $Cm$ and LB + $Kan$ + $Cm$ plates, and the plates were incubated at 30 °C overnight to form lysogen colonies. After lysogenic growth, colonies were counted to determine pure and mixed integrations. This assay to determine integrations was done in both LB and M9 media, where the M9 experiment differs from the LB method by using M9M in place of LBMM, and M9 + 0.4% glucose (M9G) in the place of LBM and LBGM when using liquid media. Plating is done on LB agar plates.

Mixed integrations are defined as lysogens that grew on the LB + $Kan$ + $Cm$ plates:

$$\text{MixedIntegrations} = \text{Lysogens}_{Kan+Cm}$$

Pure $Kan^R$ integrations are defined as:

$$\text{PureIntegrations}_{Kan} = \text{Lysogens}_{Kan}\text{Lysogens}_{Kan+Cm}$$

Which are the lysogens growing on the LB + $Kan$ plates subtracted by the mixed integrations, and pure $Cm^R$ integrations are defined in the same manner but using the LB + $Cm$ plates:

$$\text{PureIntegrations}_{Cm} = \text{Lysogens}_{Cm}\text{Lysogens}_{Kan+Cm}$$

The total lysogens are defined as the sum of the mixed integrations and both pure integrations. The frequencies were shown in Fig. 4a.

Similarly for the complementation lysogenization assay, a 1:1 mixture of WT phages bearing a $Cm^R$ marker and mutant phages (λ$cII^-$ or λ$P^-$) bearing a $Kan^R$ marker (λWT-$Cm$ + λ $cII^-$-$Kan$ or λ $P^-$-$Kan$) was used for the infection. The plating step was as described above on different antibiotic plates. The total WT integrations for mixed-phage infections are defined as the sum of the mixed integrations and pure WT integrations (mixed integrations + pure integrations$_{Cm}$ as calculated above) and the total mutant integrations for mixed-phage infections is calculated in a similar manner (mixed integrations + pure integrations$_{Kan}$ as calculated above). Also, for the double mutant complementation, the λ $cII^-$ phage with a $Kan^R$ marker was mixed 1:1 with the λ$P^-$ phage with the $Cm^R$ marker (λ$cII^-$-$Cm$ + λ$P^-$-$Kan$). The total integrations for each mutant are calculated in the same manner as the WT and mutant above.

**Bulk mixed lysis assay and calculations.** Host cells MG1655[pBR322-PLate*D] were grown in the same manner as described for the movies (in M9M + $Amp$). One-hundred fifty microlitres of the same phage mixture as used for the movies was mixed with 150 μl of the resuspended host cells and kept on ice for 30 min. The infection mixture was then moved to a 42 °C water bath for 5 min, then the mixture was diluted into 3 ml of fresh M9G, and moved back to the 42 °C water bath for 10 min, then moved to a 37 °C shaking water bath until visible lysis occurred. This procedure ensures that all infected cells lyse to produce more phages for analysis. Following lysis, the lysate was moved to a tube and chloroform (Fisher Scientific) was added to reach 5% volume of the mixture and mixed with a tube shaker (Fisher Scientific) for 10 min, then centrifuged (4 °C, ~4,800g) to pellet debris, and the supernatant was transferred to a new tube. This chloroform and pelleting process was repeated twice more. The lysate was imaged under the microscope using the lytic reporter channels to determine pure blue, pure green and mixed phage progeny. This mixed lysis experiment was done in both M9 and LB media, where the LB version differs from the M9 method by replacing M9M with LBMM, and replacing M9G with LBM.

The same phage mixture as in the movie was used (API ~2) to infect cells with the pBR322-PLate*D plasmid to help generate stable phage progeny, and the predicted values are based on that API. The prediction is based on the assumption that phage adsorption follows the Poisson distribution[35], the different phages adsorb in an unbiased manner (the mixture of phages behaves uniformly, so dual-colour infections follow the binomial distribution), and that mixed lytic infections result in mixed phage progeny. The lysate was examined under the microscope for phage foci of one (pure, blue or green) or both (mixed, blue and green) colours.

At a given API, the distribution of different phage MOIs can be determined using the Poisson distribution,

$$P(M, \lambda) = \frac{\lambda^M e^{-\lambda}}{M!}$$

Where M = MOI and $\lambda$ = API, giving the population fraction of individual MOIs. Since the phage mixture is a 1:1 mixture of blue and green phages, the fraction of mixed infected cells with one or more of each phage can be determined for different MOIs using the binomial distribution,

$$G(M) = 1 - 2*\left(\frac{1}{2}\right)^M$$

where M = MOI. By combining the equations we obtain,

$$V(\lambda) = \sum_{M=1}^{\infty} P(M, \lambda) G(M)$$

Which gives the fraction of mixed infected cells as a function of the API. For API = 2, 40% of cells are expected to be mixed infected, and, therefore, ~40% of the fluorescent phage foci would be expected to have both colours, assuming that phages do not interfere with each other's lytic development.

**Computational methods.** We formulated two separate simple biochemical reaction models, one for the lytic fate and one for the lysogenic fate. These were both simulated using a stochastic simulation, the tau-leap[58], an offspring of the Gillespie stochastic simulation algorithm[59]. Detailed descriptions can be found in the Supplementary Methods.

**Data availability.** All relevant data and Matlab codes are available from the authors upon request.

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

## Acknowledgements

We are grateful to Ryland Young, Craig Kaplan and Michael Cortes for commenting on the earlier versions of the manuscript. We thank Ryland Young, Alan Davidson, Theodorus W.J. Gadella and Stanislav Vitha for gifting strains, and Luis Rene Garcia for his dissecting microscope to screen fluorescent plaques. This work was supported by the National Institutes of Health (R01GM107597 to J.T.T., T.S., Q.S., G.B and L.Z.), TAMU-NSFC grant from Texas A&M University (02-230242 to J.T.T.) and the Louis and Beatrice Laufer Center for Physical and Quantitative Biology to G.B.

## Author contributions

J.T.T. and L.Z. designed the experiments. J.T.T. conducted the experiments. J.T.T. and Q.S. analysed the data. T.S., J.T.T. and G.B. built the computational models. J.T.T., T.S., G.B. and L.Z. wrote the paper. L.Z. supervised the project.

## Additional information

**Competing financial interests**: The authors declare no competing financial interests.

**How to cite this article**: Trinh, J. T. *et al.* Cell fate decisions emerge as phages cooperate or compete inside their host. *Nat. Commun.* **8**, 14341 doi: 10.1038/ncomms14341 (2017).

**Publisher's note**: 

