## [Peer Review File · Nature Communications]

Reviewers' comments:

Reviewer #1 (Remarks to the Author):

Review of Trinh et al. for Nature Communications

The iconic bacteriophage lambda-E. coli model system has provided significant insight into cellular processes. Despite being one of the best studied gene regulation systems, much is still unknown about phage lambda intracellular behavior, particularly with respect to how phage DNA is processed. Trinh et al. analyze phage lambda infection at the highest resolution yet with their 4 color reporter system. Their results are surprising, indicating that an early infecting phage can dominate a later infecting phage in the lytic life cycle, but cooperate in the lysogenic cycle. The supporting data are convincing, and the arguments are presented cogently and concisely in this well written manuscript. I fully support its publication in Nature Communications.

L78: I'm assuming the dots in Fig. 1 panels b, d and f are bound phage. If so, why does lysis take so long? Under normal conditions, phage lambda lyse in ~60 min.

L91-92: I don't think you should be adding the adjective "minor" here because that might be debatable. I'm not an expert in fluorescent imaging, and I didn't completely follow the arguments presented in the text and in the supplementary figure. Are there citations that give a more systematic justification of the adjustments made to the data? The yellow-red crosstalk is concerning since the implication is that the relative contributions of the coinfecting phages may be conflated. Likewise the green-yellow crosstalk is concerning because lysis/lysogeny proteins from different phages might be conflated. Does the level of crosstalk differ between single and multiple infections? Does crosstalk vary with changing intensity, and how does this affect the rescaling of intensity values?

L105-106: The suggestion of competitive interaction between coinfecting phages whereby one phage is able to prevent the expression of the other's genes by arriving earlier is probably the most surprising, and most controversial, claim the paper makes. I was fully prepared to offer an objection to this finding, but the authors have thoroughly considered the alternative and have presented a convincing case.

L121...somewhere. I think it would be worthwhile to point out that, and I believe the data show this, the frequency of mixed infections 'won' by green = frequency 'won' by blue. This evidence demonstrates that nothing intrinsically different about green vs. blue, but rather it is likely timing of DNA entry into cell determines winner. In other words, winning is random.

L341: Do you mean Alan Davidson?

L443: Strictly speaking this is probably not true. See Gallet et al. 2012 in Evolution. That said, I am not sure it impacts your claim.

Reviewer #2 (Remarks to the Author):

In the work by Trinh et al., the authors developed a four-color fluorescent reporter system to investigate how multiple λ phages infected on the same cell decide to either lyse or integrate into the host chromosome to become a lysogen. Using a combination of single cell imaging, bulk assays, and computational modeling, the authors showed that multiple phages interact within the same cell — they cooperate during lysogeny, but compete during lysis. The authors further investigated possible mechanisms underlying these phenomena and showed that the competition between phage in the lytic

state is mainly due to differences in the cellular arrival time of different phage DNAs and limited resources available for a high level of phage DNA replication. Establishment of lysogen does not require a high level of resources, and phages often collaborate to integrate into the host genome. The phenomenon of multiple phages competing or collaborating in the same cell serves as a simplified case for how cells make a fate-determining decision, which is of particular importance in development and differentiation of higher organisms. The data presented in the manuscript supports the major conclusions; appropriate controls were performed, and both the biological insight learned in this study and also the new imaging method will be of interest to a wide scientific community. However, the manuscript can be improved by clarifying the descriptions of some experiments, in particular, the bulk infection assays, which are important to support the computational models. I will support the acceptance of the manuscript provided the authors improve their languages of the manuscript and address my concerns below.

1. How do the differences in the maturation time of different fluorescent proteins affect the determination of assigning phage fates? mKO and mKate mature slowly, on a similar or longer time scale compared to the experimental time window (160 – 240 min).
2. Is there any crosstalk between the blue channel and the other channels? In Figure 6a, top row, it appears that similar fluorescence patterns were observed in green, blue and yellow channels, most likely due to crosstalk between these channels. How does this affect the quantification of the mixed voting? Are those images already corrected for bleed-through according to the methods section? This would be of serious concern if the degree of crosstalk is high. In addition, the yellow fluorescence image appears to be clustered in the cytoplasm, which should not be.
3. Does continuous laser illumination, in particular, blue light, induce SOS response of the host cell, which would affect the outcome?
4. The description in the main text regarding the bulk lysogenization assay (p8, 2nd paragraph) is not clear. There is no description of how this assay was done, and in the corresponding figure legend (Fig. 3h) it appears there are phages of different antibiotic resistance, which is not mentioned in the main text. Similarly, the 3rd paragraph on the same page lacks sufficient details about the assay. These are important experiments and need to be clearly described.
5. Please provide more details in method regarding the imaging procedure: were images of the four color channels acquired simultaneously or sequentially?

Figure Comments:

1. Figure 3 b, c, e, f it appears the labels are reversed between the two panels. The top panel should be lytic signal and the bottom panel should be lysogenic signal.
2. In Figure 4 some of the bars have error bars while some don't. Please also provide the sample size for the calculation of these error bars.
2. Figure 5 the white bars in the stacked columns are not labeled as others.

Detailed Responses to Reviewers' Comments

Reviewer #1

We would like to express our appreciation for Reviewer #1's careful reading of the manuscript. We are very happy that this reviewer considered our results to be "surprising," and found that our "arguments (were) presented cogently and concisely" in the manuscript. This reviewer "fully supports" this manuscript for publication and for that, we are very thankful.

The reviewer had a few questions/concerns which we will do our best to address here.

1: "L78: I'm assuming the dots in Fig. 1 panels b, d and f are bound phage. If so, why does lysis take so long? Under normal conditions, phage lambda lyse in ~60 min."

Yes, the dots, pointed to by the symbols, are indeed the adsorbed phages. It is true that the lysis time in our study here is long on average, ~115min (Figure 6c). The ~60 min that is being referred to however, is from studies using LB as the growth medium, and with the cells/phages in culture flasks with good aeration (Wang, I.N., Smith, D.L., and Young, R. *Annu Rev Microbiol.* 2000.; Shao, Y. and Wang, I.N. *Genetics.* 2008). On the other hand, in our study we use M9 as the growth medium, and the cells/phages are under an agarose pad and imaged under the microscope without shaking. Under conditions similar to these, lysis times have been observed to be comparable with the lysis times we observe (Zeng, L. et al. *Cell.* 2010.; St-Pierre, F. and Endy, D. *PNAS.* 2008.).

2a: "L91-92: I don't think you should be adding the adjective "minor" here because that might be debatable."

We have removed "minor" from line 92.

2b: "Are there citations that give a more systematic justification of the adjustments made to the data?"

Our method of correcting is based on a standard way that people correct for crosstalk, for example in studies using FRET, where they have controls without the FRET acceptor to determine the crosstalk leakage, and subsequently subtract a fixed fraction from their observed signals (Park, J., and Ha, T. *Imaging: A Laboratory Manual.* 2011). For our experiments, our method is a logical one to determine crosstalk. By doing an actual pure phage infection, identically to the mixed infections, we can determine the expected fluorescence levels of our actual reporters and therefore quantify our expected range of crosstalk, as to not under-correct.

2c: "The yellow-red crosstalk is concerning since the implication is that the relative contributions of the coinfecting phages may be conflated. Likewise the green-yellow crosstalk is concerning because lysis/lysogeny proteins from different phages might be conflated."

We agree that any crosstalk has the potential to obfuscate our cell fate assignments, and fully understand your concern. This is why we performed the crosstalk controls and quantification. We did infection movies with only the "blue phage," which expresses only the yellow lysogenic reporter and observed some apparent red signal, and found the crosstalk level to be ~5% (Supplementary Fig. 2e). A very high level of yellow

signal needs to be achieved to give a fair level red crosstalk (this is prior to the scaling as well). Our yellow signals in lysogens (pure and mixed, completely unaffected by crosstalk) have a mean of $\sim 10x$ in our experiments, meaning that the average crosstalk (yellow to red) is $\sim 0.5x$. As a comparison, our pure red lysogens have a mean signal of $\sim 9x$ background (no possible crosstalk). In conclusion, what we classify as red lysogens have signals that are high enough to be unambiguously classified as a true signal, as opposed to crosstalk.

The green-yellow crosstalk is similar, where we observed $\sim 6\%$ crosstalk (using only the “green phage” for movies). The number of cells we classified as actually having both yellow and green signals is low, and the average green signal in those cells is about $10x$ background.

Another observation about the green-yellow crosstalk is that green is the lytic reporter, and as such, it tends to form distinct foci, which would also exist in the yellow channel in the case of crosstalk. When assigning cell fates, it is clear that cells we assigned as having both yellow and green signals had yellow fluorescence throughout the cell and not just in foci, which we interpret as true signal, since the levels of green signal are not high enough to make crosstalk signal evenly disperse throughout the cell where there are no foci. **We have added a description of how the lytic reporter forms foci in the main text in lines 80–82.**

Additionally, we can observe the yellow signal throughout the infection, including before the green signals (potential crosstalk) become clear and bright. Since there is a uniform low level of yellow signal in the absence of green, it cannot be crosstalk.

2d: “Does the level of crosstalk differ between single and multiple infections?”

No, they do not. The level of crosstalk varies only with the intensities of the signals.

2e: “Does crosstalk vary with changing intensity, and how does this affect the rescaling of intensity values?”

Yes, they vary with the intensity, which we quantified the percentage of crosstalk signal in Supplementary Fig. 2d and e. It does not actually affect the scaling because we subtracted the crosstalk before scaling the signals. Also, we normalized our signals to the maximum of each channel, and those values were unaffected by crosstalk as they came from cells with only a single signal.

3: “L105-106: The suggestion of competitive interaction between coinfecting phages whereby one phage is able to prevent the expression of the other’s genes by arriving earlier is probably the most surprising, and most controversial, claim the paper makes. I was fully prepared to offer an objection to this finding, but the authors have thoroughly considered the alternative and have presented a convincing case.”

We are glad that we convinced you with our explanation.

4: “L121...somewhere. I think it would be worthwhile to point out that, and I believe the data show this, the frequency of mixed infections “won” by green = frequency “won” by blue. This evidence demonstrates that nothing intrinsically different about green vs. blue, but rather it is likely timing of DNA entry into cell determines winner. In other words, winning is random.”

This comment offers a good suggestion, and the data do indeed support this. Within the mixed infected population, blue:green “wins” = 45:41, which is practically tied. **We have added a comment to this effect in lines 108-109.**

5: “L341: Do you mean Alan Davidson?”

No, this line refers to Michael Davidson, who has graciously deposited plasmids with fluorescent proteins to Addgene, which we used. Alan Davidson did contribute to our work by gifting us some phages (Supplementary Table 2), so that may be the source of confusion.

6: “L443: Strictly speaking this is probably not true. See Gallet et al. 2012 in *Evolution*. That said, I am not sure it impacts your claim.”

Traditionally, people have made this assumption of Poisson distribution for phage adsorption, when studying phage lambda (Kourilsky, P. *Mol Gen Genet.* 1973) as well as other phages (Benzer, S. *J Bacteriol.* 1952.). Per our understanding, in Gallet et al. *Evolution* 2012, they are studying the kinetics of adsorption, they find that the rate of adsorption varies within the phage population. The distributions that they assume are not directly applicable to our calculations, as we use a standard protocol to maximize the phage adsorption by allowing the process to occur on ice for a period of time without DNA ejection (Mackay, D. J., and Bode, V. C. *Virology.* 1976), therefore we are observing the end result of adsorption, which will have a distribution. In addition, we have examined our fluorescently labeled phages and determined how they adsorb under our conditions with single-phage resolution under the microscope, which does follow the Poisson distribution across multiple APIs, consistent with our previous studies (unpublished data from the work of Zeng, L., et al. *Cell.* 2010.). We feel that it is legitimate to make this adsorption assumption in order to calculate the specific value in the manuscript.

In the figure that includes this calculation, Figure 3h, our main claim is that competition in lysis is affected by resources, which we test using different growth media. We agree with you that Poisson assumptions do not affect this claim, as our observed differences are independent of any adsorption assumptions.

7 (Additional simulation comments)

We wanted to clearly state how we changed our computational modeling during revision. We made a single parameter change to the lytic protein production rate to better agree with published final protein values (lytic model: k4). Due to this, we re-ran simulations to make new representative figure panels in Figure 2e, Figures 3a-f, and Supplementary Figure 4, which may look different due to the stochastic nature of the simulations. We used different ranges for the x and y axes to efficiently use plot space. The simulations were re-run with identical settings outside of the changes we stated.

8 (Additional figure comments)

We removed the scale bar text from all microscopy images following the journal’s guidelines. We also changed the color scheme and stack ordering in the bar graphs in Figure 2a, 3g and h, to keep them consistent with each other.

Reviewer #2:

We would like to thank Reviewer #2 for reading our manuscript and for the insightful comments which raise important points. We are happy that this reviewer thought that our data “supports the major conclusions” and that our methods and results “will be of interest to a wide scientific community.” This reviewer “will support the acceptance of the manuscript” assuming we address the points raised, and for that, we are thankful and will do our best to satisfy any concerns here.

1: “How do the differences in the maturation time of different fluorescent proteins affect the determination of assigning phage fates? mKO and mKate mature slowly, on a similar or longer time scale compared to the experimental time window (160 – 240 min).”

This comment raises an important point about being able to detect the fluorescence in the time of the experiment. The published maturation half-times of the proteins we use, mKO2 and mKate2, in *E.coli* at 37 °C, are about 72 and 20 minutes respectively (Strack, R.L., et al. *BMC Biotechnology*. 2009.; Shcherbo, D., et al. *Biochem J*. 2009), which are well within the bounds of our movies. The mKO and mKate proteins are earlier variants of the ones we use with slower maturation kinetics. In our movies, we observe that fluorescence appears during the movies and gets brighter presumably as the FP matures (Movie S1; Movie S2). We actually let the movies run to around 4 hours or more, which is sufficient time to observe the fluorescence in lysogenic cells, as well as all of their progeny, as they divide multiple times during the course of the movies. Although probably not fully mature, we can still see the mKO2 and mKate2 fluorescence in our mixed voting cells which lyse relatively early on, so detection and subsequent assignment of fates was not of concern.

2a: “Is there any crosstalk between the blue channel and the other channels?”

This is keen observation, since our “blue” protein’s fluorescence is fairly close to the green channel. There is no crosstalk of blue into any other channel, under our imaging conditions. We would like to point out that we are using a custom filter set for capturing green fluorescence (Methods), which is different than a standard GFP cube. It is narrower in its excitation bandwidth, which should help prevent blue crosstalk. Specifically, our “blue” protein is a cyan FP and our “green” protein is a green-yellow FP, so their fluorescence profiles have more separation than traditional CFP and GFP, as they are closer to how CFP and YFP are separated. CFP and YFP make up an oft-used pair that is well characterized to be compatible with each other.

2b: “In Figure 6a, top row, it appears that similar fluorescence patterns were observed in green, blue and yellow channels, most likely due to crosstalk between these channels. How does this affect the quantification of the mixed voting.”

This is also a good observation, and it has a simple explanation based on how the phages assemble during lysis. In that panel we depict a “cross-phage mixed vote,” so both phages lyse and lysogenize. For all mixed lysis events where both blue and green are observed, there is always blue/green co-localization (Fig 1f; Supplementary Fig. 2a). This is not due to crosstalk, because there is no detectable blue or green crosstalk in

either direction under our conditions. We speculate however that if both phages are lysing, then there will be two pools of gpD fusion proteins, each with a different FP and phages will assemble randomly with both. Therefore the co-localization observed, usually seen as foci, is probably due to phage assembly in those regions of the cell. **We have added a description of this in the main text in lines 80-82.** There is the green-yellow crosstalk though, which we discussed in the main text.

There is no crosstalk effect on classifying mixed voting cells. For the “same-phage type mixed votes,” the color pairs (blue/yellow and green/red) have no crosstalk. For the “cross-phage mixed votes,” there are three subtypes (blue/green/red, blue/green/yellow, and blue/green/red/yellow), where green and yellow can coexist. An important note about green-yellow crosstalk is that green lytic signals form foci, which in turn form crosstalk yellow foci. The cells classified as mixed voting cells also have yellow signal throughout the cell, as they express the fluorescence where there should not be crosstalk as well. Therefore if the cell has significant yellow signal in non-crosstalk areas, we can confidently assign its fate.

2c: “Are those images already corrected for bleed-through according to the methods section? This would be of serious concern if the degree of crosstalk is high”

The actual images are not manipulated to remove crosstalk. The Methods section is stating that we correct the actual quantified signals as appropriate for crosstalk, during our analysis. We understand your concern about the level of crosstalk, since it would be near impossible to distinguish between certain color combinations if the level was too high.

This is why we performed the crosstalk controls and quantification. We did infection movies with only the “blue phage,” which expresses only the yellow lysogenic reporter and observed some apparent red signal, and found the crosstalk level to be ~5% (Supplementary Fig. 2e). A very high level of yellow signal needs to be achieved to give a fair level red crosstalk. Our yellow signals in lysogens (pure and mixed, completely unaffected by crosstalk) have a mean of ~10x in our experiments, meaning that the average crosstalk (yellow to red) is ~0.5x. As a comparison, our pure red lysogens have a mean signal of ~9x background (no possible crosstalk). We were fortunate that we did not have a situation where one reporter was expected to be exceptionally high, and the other expected to be low. For us, the fluorescence levels of both of these channels were fairly comparable, so the effect of crosstalk is diminished. In conclusion, what we classify as red lysogens have signals that are high enough to be unambiguously classified as a true signal, as opposed to crosstalk.

The green-yellow crosstalk is similar, where we observed ~6% crosstalk (using only the “green phage” for movies). This is only relevant in cells we classified as actually having both yellow and green signals, which is low, and the average green signal in those cells is about 10x background. Again, the green-yellow crosstalk features foci due to the lytic reporter, and as such, it tends to form distinct foci, which would also exist in the yellow channel in the case of crosstalk. When assigning cell fates, it is clear that cells we assigned as having both yellow and green signals (mixed voting) had yellow fluorescence throughout the cell and not just in foci, which we interpret as true signal, since the levels of crosstalk are not high enough to evenly disperse throughout the cell where there are no foci. Additionally, we can observe the yellow signal throughout the

infection, including before the green signal (potential crosstalk) become clear and bright. Since there is a uniform low level of yellow signal in the absence of green, it cannot be crosstalk.

Finally, we would like to direct your attention to Supplementary Movie 4, from which Fig. 6a was taken from, to illustrate our points. We used specific and uniform contrast settings for each color in each specific figure panel with microscopy images or movies, meaning if something appears to get brighter, then it actually is, although the pixel density saturates in places. Comparing panels a and c in the green channel, we can see that they are similar, but in their yellow channels, they are different. The crosstalk from green to yellow in panel c is clearly not strong enough to elicit a visible signal, whereas under the same contrast in panel a, we see yellow signal in the cell. We interpret this as actual signal in panel a, which is amplified in certain areas due to crosstalk, but we can see that the yellow signal is also where the green is not. We can also see that in panels a and b of the movie, the yellow channel does not have consistent clumping (disregarding the green crosstalk), and the signals change with each frame of the movie. So this apparent localization is just an imaging artifact which is transient, as opposed to actual localization in the lytic channels which are persistent.

2d: “In addition, the yellow fluorescence image appears to be clustered in the cytoplasm, which should not be.”

We feel like we have addressed part of this concern in the response above because of the green-yellow crosstalk. For the other areas of the cell where there is apparent “clumping” we think it may be due to the image processing. For Fig 6a in particular, we wanted to emphasize the colors, so we uniformly adjusted the contrast of that channel for the whole panel to make the yellow very bright. In comparison, in Fig. 1d with a different contrast setting, we see no clumping.

3: “Does continuous laser illumination, in particular, blue light, induce SOS response of the host cell, which would affect the outcome?”

This is an insightful comment regarding the biology of the phage and cell, as many temperate phages would have their lysogenic cycles disrupted by the host SOS response. Firstly, our imaging is not continuous laser illumination. We use wide field illumination from an X-Cite 200DC mercury lamp and expose the cells intermittently over time, thereby minimizing the intensity of light the cells are exposed to. Specifically for the blue light, the cells are exposed to 9 z-stacks of 200ms exposures at the beginning of the movie to locate the phages around the cell, and following that, they receive a 40ms exposure every 5 minutes. **We have added these additional details within our Methods in lines 399-412.** Secondly, we have done controls to show that the overall growth rate of the cells and cell morphology are not affected by this level of exposure. Thirdly, the phages we use for our movies bear the *cl857* allele, which is not UV inducible (Sussman, R. and Jacob, F., *CR Acad Sci.* 1962; Petranovic, M., et al. *J Bacteriol.* 1979). Lastly, we do not observe frequent filamentation in our cells (~1% of uninfected cells), indicative of DNA damage/SOS response (Little, J.W., and Mount, D.W. *Cell.* 1982), so we do not believe it to be a concern.

4: “The description in the main text regarding the bulk lysogenization assay (p8, 2nd paragraph) is not clear. There is no description of how this assay was done, and in the corresponding figure legend (Fig. 3h) it appears there are phages of different antibiotic resistance, which is not mentioned in the main text. Similarly, the 3rd

paragraph on the same page lacks sufficient details about the assay. These are important experiments and need to be clearly described.”

We think that all of these assays were detailed appropriately in the Methods, but for additional clarity we rewrote part of that paragraph to be more descriptive about the assay in the main text in lines 174-178. As for the figure legend for Fig. 3h, we were not clear in the description, but those phages were the exact phages that we used for the movies, so they have the same antibiotic resistance (chloramphenicol). We have added in those details in line 183-185 and in the figure legend in lines 746-747. In the following paragraph, for the mutant mixing lysogenization assay, we have also rewritten parts of it to be more clear and descriptive of the assay in lines 192-198. We also made a comment pointing out that the antibiotic resistance of the phages has no observed effects on the decision-making behavior (lines 200-201).

5: “Please provide more details in method regarding the imaging procedure: were images of the four color channels acquired simultaneously or sequentially?”

We agree that it would be valuable to have this information for reproducibility and detail. In our setup we set up the imaging parameters for each stage, and choose stages to image, which are then carried out by the computer, and the imaging for each channel is all sequential using filter cubes. The program will run through all of channels for a stage and then move on to the next stage and repeat. Each movie is done in two parts. In the first part, we image z-stacks to locate the phages, so we capture the blue and then green channels for each z-plane. On the focal plane of this first part, the capture order is: phase-contrast, red, yellow, blue, green. For the second part of the movie, the time-lapse portion, the capture order is the same: phase-contrast, red, yellow, blue, green. We have added these additional details in our Methods in lines 399-412.

6 (Figure comment 1): “Figure 3 b, c, e, f it appears the labels are reversed between the two panels. The top panel should be lytic signal and the bottom panel should be lysogenic signal.”

The labels were actually correct, but this has raised an important issue with our presentation of the histograms. We have updated Figure 3, by changing the color scheme and the specific way the data are plotted to help give a clearer portrayal of the data. We also have included a more detailed description in the legend for this figure.

We changed the color scheme of Figure 3a and d, and the histograms (Figure 3 b, c, e, f) to that of Figure 2b, c, e, f, where lytic data are in turquoise and lysogenic data are in orange. We also changed the range of our x and y axes to use the plot space more efficiently. These normalized data in the histograms are divided into 20 bins, and we are now showing bins 1-12, which shows the overall trend of the data while also cutting extraneous white space (bins 13-20). The full plots are included in Supplementary Figure 4. We added threshold lines that separate the pure outcomes (i.e. < 5% signal in one reporter and > 5% signal in the other) from mixed (> 5% in both reporters). The data in the bottom left bin (< 5% in both reporters) are counted as dead cells and excluded from the pure/mixed percentages (< 0.3% of the data in all cases are in this bin). In the previous version of this figure, we showed only bins 2-20, which purposely did not show the pure signals (although the full data was also presented in Supplementary Figure 4).

We hope that this version of the panels is clearer in showing that the signal density shifts from mostly pure to mostly mixed as resources are increased (and vice versa as delay is increased) in the lytic histograms. In contrast, the lysogenic signal density stays mostly within the mixed region. In constructing the new figures, we also ran the simulations again under the same delay and resource settings as before.

7 (Figure comment 2): “In Figure 4 some of the bars have error bars while some don't. Please also provide the sample size for the calculation of these error bars.”

All of the points and bars in Fig. 4 have error bars in actuality, but many are within the symbols and also are too small to see on the bars, at least in these lower resolution figures, but they are all visible in the separate figure pdfs. The error bars are standard deviations from two technical replicates in the scatter plots, and those errors were propagated to the bar graphs. The data shown are from a representative biological replicate, containing two technical replicates, and at least 2 biological replicates were done for each experiment. **We have added a description of the replicates we performed in the figure legends in lines 758-760.**

8 (Figure comment 3): “Figure 5 the white bars in the stacked columns are not labeled as others.”

On our end, in the pdf, there seems to be no problem, both the black and white stacked bars have numbers contained within them. Hopefully this is not a problem in the end. Also it may be an issue with printing straight from the email.

9 (Additional simulation comments)

We wanted to clearly state how we changed our computational modeling during revision. We made a single parameter change to the lytic protein production rate to better agree with published final protein values (lytic model: k4). Due to this, we re-ran simulations to make new representative figure panels in Figure 2e, Figures 3a-f, and Supplementary Figure 4, which may look different due to the stochastic nature of the simulations. The simulations were re-run with identical settings outside of the changes we stated.

10 (Additional figure comments)

We removed the scale bar text from all microscopy images following the journal's guidelines. We also changed the color scheme and stack ordering in the bar graphs in Figure 2a, 3g and h, to keep them consistent with each other.

REVIEWERS' COMMENTS:

Reviewer #1 (Remarks to the Author):

I am satisfied that the authors have addressed all concerns, and congratulate them on an interesting and excellent work.

Reviewer #2 (Remarks to the Author):

In the revised work by Trinh et al., the authors have addressed all of the concerns raised previously. I believe the work is fit for publication and will be of interest to a wide scientific community.

Detailed Responses to Reviewers' Comments

Reviewer #1 (Remarks to the Author): I am satisfied that the authors have addressed all concerns, and congratulate them on an interesting and excellent work.

We are happy to see that the reviewer is satisfied with our changes and find the manuscript fit for publication.

Reviewer #2 (Remarks to the Author): In the revised work by Trinh et al., the authors have addressed all of the concerns raised previously. I believe the work is fit for publication and will be of interest to a wide scientific community.

We are happy to see that the reviewer is satisfied with our changes and find the manuscript fit for publication.